# Escalating spread of SARS-CoV-2 infection after school reopening among students in hotspot districts of Oromia Region in Ethiopia: Longitudinal study

**Dabesa Gobena** [1,2]☯*, **Esayas Kebede Gudina**[3]☯, **Daniel Yilma**[3], **Tsinuel Girma**[4], **Getu Gebre**[1], **Tesfaye Gelanew**[5], **Alemseged Abdissa**[5], **Daba Mulleta**[1], **Tarekegn Sarbessa**[1], **Henok Asefa**[6], **Mirkuzie Woldie**[4], **Gemechu Shumi**[4], **Birhanu Kenate**[4], **Arne Kroidl**[7,8], **Andreas Wieser**[7,8], **Beza Eshetu**[9], **Tizta Tilahun Degfie**[4]☯, **Zeleke Mekonnen**[2]☯

1 Public Health Emergency Management and Health Research Directorate, Oromia Health Bureau, Addis Ababa, Ethiopia, 2 School of Medical Laboratory Science, Institute of Health, Jimma University, Jimma, Ethiopia, 3 Department of Internal Medicine, Jimma University, Jimma, Ethiopia, 4 Fenot Project, School of Population and Public Health, University of British Columbia, Addis Ababa, Ethiopia, 5 Armauer Hanssen Research Institute, Addis Ababa, Ethiopia, 6 Department of Epidemiology, Jimma University, Jimma, Ethiopia, 7 Division of Infectious Diseases and Tropical Medicine, Medical Center of the University of Munich, Munich, Germany, 8 German Center for Infection Research, Partner Site Munich, Munich, Germany, 9 Department of Pediatrics and Child Health, Jimma University, Jimma, Ethiopia

☯ These authors contributed equally to this work.
* dabesagobena25@gmail.com

**Data Availability Statement:** All relevant data are within the paper and its Supporting information files.

## Abstract

### Background

COVID-19 pandemic caused by extended variants of SARS-CoV-2 has infected more than 350 million people, resulting in over 5.5 million deaths globally. However, the actual burden of the pandemic in Africa, particularly among children, remains largely unknown. We aimed to assess the seroepidemiological changes of SARS-CoV-2 infection after school reopening among school children in Oromia, Ethiopia.

### Methods

A prospective cohort study involving students aged 10 years and older were used. A serological survey was performed twice, at school reopening in December 2020 and four months later in April 2021. Participants were selected from 60 schools located in 15 COVID-19 hotspot districts in Oromia Region. Serology tests were performed by Elecsys anti-SARS-CoV-2 nucleocapsid assay. Data were collected using CSentry CSProData Entry 7.2.1 and exported to STATA version 14.2 for data cleaning and analysis.

### Results

A total of 1884 students were recruited at baseline, and 1271 completed the follow-up. SARS-CoV-2 seroprevalence almost doubled in four months from 25.7% at baseline to

**Funding:** There is no specific funding for this study. In collaboration with different universities in the region and a partner (Fenot project) team, Oromia Health Bureau initiated this work to strengthen COVID-19 surveillance and innovative approach to COVID-19. The collaborative team develops the proposal as voluntary work and resources pulled from different organizations as part of evidence generation for the routine activity of fighting COVID-19. The team organized the workshop during data analysis and arranged the transport for the research team to supervise the testing procedures at the Adama reference laboratory (this laboratory is under the management of the Oromia Health Bureau). A platform was established to bring programmers (Oromia Health bureau) and researchers from Jimma University and Armauer Hansen Research Institute. It is to provide continuous technical support on serological testing, and Jimma University provides Elecsys Anti-SARS-CoV-2 immunoassay test kits). Overall, this study is an effort from different sectors to generate evidence to support decision-making in fighting COVID-19 in the Oromia region, Ethiopia. Hence, no specific funding was allocated for this study. The funders had no role in the study design, data collection, analysis, publication decision, or manuscript preparation.

**Competing interests:** The authors have declared that no competing interests exist.

46.3% in the second round, with a corresponding seroincidence of 1910 per 100,000 person-week. Seroincidence was found to be higher among secondary school students (grade 9–12) compared to primary school students (grade 4–8) (RR = 1.6, 95% CI 1.21–2.22) and among those with large family size (> = 5) than those with a family size of <3 (RR = 2.1, 95% CI 1.09–4.17). The increase in SARS-CoV-2 seroprevalence among the students corresponded with Ethiopia's second wave of the COVID-19 outbreak.

## Conclusion

SARS-CoV-2 seroprevalence among students in hotspot districts of the Oromia Region was high even at baseline and almost doubled within four months of school recommencement. The high seroincidence coincided with the second wave of the COVID-19 outbreak in Ethiopia, indicating a possible contribution to school opening for the new outbreak wave.

## Research in context

### Evidence before this study

SARS-CoV-2 infection remains a global health priority. There is limited evidence about the burden of the disease on school children and their role in outbreak propagation. In Africa, where most countries preemptively closed schools at the beginning of the pandemic, the disease burden on children and the impact of school closure or opening on the national outbreak pattern has never been studied. We did electronic searches of MEDLINE, EMBASE, medRxiv, and Pubmed for articles published in English from December 2019 to December 2020 using the terms "COVID-19" OR "SARS-CoV-2" AND "School" OR "School children.". Our search yielded only a few studies conducted in Europe and the United States of America; none from Africa. These studies showed that school closure had helped countries to avoid a surge of cases. However, school children suffered greatly from the lockdown and other public health measures than the pandemic. This made countries make a difficult choice of whether to return children to school or strengthen public health measures, including keeping the schools closed to contain the pandemic. The available resource indicated that a SARS-COV-2 sero survey study would support the country in estimating the status of the pandemic, monitoring and evaluating the implementation status of COVID-19 preventive and control measures [37–48]

### Added value of this study

To our knowledge, this is the first serological study on COVID-19 among school-age children in Ethiopia. This longitudinal follow-up study examined the antiSARS-CoV-2 infection seroprevalence and seroincidence among school-age students in the Ethiopian context. We used a high sensitivity and specific serological test (Elecsys® Anti-SARS-CoV-2, Roche Diagnostics) to study SARS-CoV-2 serology. In this study, we investigated the seroincidence change after school recommencement and tried to assess factors associated with seroincidence change over time. We also tried to correlate the seroincidence change with the COVID-19 outbreak pattern in Ethiopia.

### Implications of all of the available evidence

This study has revealed that SARS-CoV-2 has been spreading steadily among school children in Oromia (Ethiopia), even during the school closure. Based on the national outbreak pattern

at the school reopening and four months later, we can speculate that school opening has contributed to the escalation of the COVID-19 outbreak in Ethiopia. However, it is also possible that the change in the outbreak pattern may also be due to the occurrence of new virus variants, as we observed with Alpha and Delta variants in 2021 and recently with Omicron [49–54]. New variants are often highly transmissible and likely to evade vaccine-mediated and natural immunity. School opening without proper public health measures creates a fertile environment for the evolution and propagation of new variants. Our data highlights the role of schools and school children in outbreak propagation and provide critical insight for future public health measures in the face of a similar outbreak.

## Introduction

Within two years of the official report from Wuhan, China, the COVID-19 pandemic caused by extended variants of SARS-CoV-2 has led to more than 350 million infections and over 5.5 million deaths globally [1, 2]. By the end of 2021, Ethiopia reported more than 405,745 cases and 6,911 COVID-19-attributable deaths. Oromia, Ethiopia's most populous regional state, reported 50,129 cases and 1,072 deaths, 12.4% and 15.5% of the nationally reported cases and deaths, respectively [3]. Although the calamities due to the pandemic are still massive, the real burden of the pandemic in Oromia, particularly Ethiopia, remains unknown.

Ethiopia preemptively closed all schools and universities after detecting the first COVID-19 case in March 2020 [4]. Such drastic measures were assumed to curtail the outbreak's spread and safeguard children and their families from the risk of COVID-19 [5]. Although such public health measures were entirely justifiable and have become a global trend during the pandemic, a tremendous impact on the learning and social life of the students was reported [6, 7]. With an ongoing outbreak, reopening schools can never be easy because the students remain susceptible when returning to the schools at a time of high community transmission [8, 9]. The risk of transmission within the school environment is expected to be high and influenced by multiple factors such as the student number per class, the room size, physical contacts during assemblage, learning sessions, and school readiness for COVID-19 mitigations [10]. The students are more exposed within the school and can be super-spreaders outside [11]. These anticipated phenomena can significantly change the dynamics and epidemiology of COVID-19 in any country if not guided by proper public health measures [9].

The real burden of the pandemic on children, in most low-income countries, in particular, remains unknown and massively underreported due to scarce laboratory testing, limited surveillance, and the milder nature of the disease in this age group [12]. As a result, the epidemiology of COVID-19 in children and their role in disease spread in countries like Ethiopia remains elusive. For instance, children <18 years of age contributed to only 4,597 (9.2%) of the 50,129 reported cases from the Oromia Region as of 28 December 2021, although about half of the Ethiopian population is younger than 20 years [13]. Thus, serological studies are the most feasible and reliable means to provide vital and robust information on key epidemiological, clinical, and virological characteristics, including improved modeling and forecasting in children [14, 15]. Serological surveys can also provide greater details regarding changes in the incidence and prevalence of a disease in a defined population [16, 17]. Most of the existing anti-SARS-CoV-2 antibody tests have very high sensitivity and specificity, detect antibodies within 1–3 weeks of symptom onset, and persist for several months. This feature will help to retrospectively assess the outbreak pattern in a given community or country [18].

Ethiopia relaxed most of the public health measures against COVID-19 in September 2020 after the end of the five-month-long state of emergency [4]. It decided to reopen schools and universities in December 2020, even though the trajectory of the outbreak was less well known

by then. By the beginning of 2021, the country started to observe a surge in cases, severe diseases, and deaths [3]. However, the impact of school commencement and relaxing COVID-19 mitigations over the outbreak pattern has never been studied. Assessing baseline serosurvey, evaluating incidence change after full-scale school reopening, and correlating this with the national outbreak pattern may help uncover the impact of school reopening on transmission dynamics. This study, therefore, aimed to report on the seroepidemiological changes and predictors among students during school closure and after reopening in hotspot districts in Oromia Region, Ethiopia.

## Methods

### Study design and settings

A longitudinal, school-based cohort study was conducted between December 2020 (baseline study) and April 2021(final cohort) in hotspot zones and towns of the Oromia Region in Ethiopia. Oromia is the largest and most populous region in Ethiopia. According to the Population and Housing Census in 2007, the region's population was projected to be 39 million by 2020, accounting for nearly 35% of the Ethiopian population; 87.7% are rural residents. Administratively, the region is divided into 21 zones, 19 town administrations, 317 districts, and 7011 *kebele*s (the smallest administrative unit in Ethiopia). The region accounts for 32% of Ethiopian land size, with 363,399.8 km$^2$.

### Selection of study participants

The study was conducted among students in selected schools located in the top 15 hotspot areas (eight zones and seven towns administrations) in Oromia Region (Fig 1).

We selected COVID-19 hotspot areas based on the number of officially reported COVID-19 cases as of 15 October 2020 (Supporting information Table 1). The schools in each town and woreda were identified and classified into four categories: public primary, private primary, public high school, and private high school. One school from the above four categories was randomly selected from each town and district. Four schools were chosen from each district, making 60 schools overall. The schools were selected and evenly distributed across all districts based on their levels (primary and high schools) and ownership (public and private). Only students of age 10 and older (grade 4 and above) were included in this study. The selected schools were stratified by grades (from grades 4 to 12), and one class from each grade was chosen randomly. It was automatically included if there was only one class in each grade. Finally, students were randomly selected from each selected class of a given grade. Then stratification by school type and grade was done to ensure adequate representation of all students who were 10 years and older. (Supporting information).

The number of participants was calculated by using single population proportion formula with the following considerations: Considering the primary objective, assuming 50% of the students already had antiSAR-CoV-2 antibodies (no prior data), the margin of error of 4%, 95% confidence interval, design effect of 3, since there were three selections (schools from each hot spot areas, class from each grade and students from each class) and by adding 10% for non-response and 10% for dropout in the second round. Finally, the sample size computed was 2164 students based on the following formula:

$$ n = \frac{Z_{\alpha/2}^{\ 2} P(1-P)}{\epsilon^2} = \frac{1.96^2 X 0.5(1-0.5)}{0.04^2} = 601*3*0.1*0.1 \cong 2164 $$

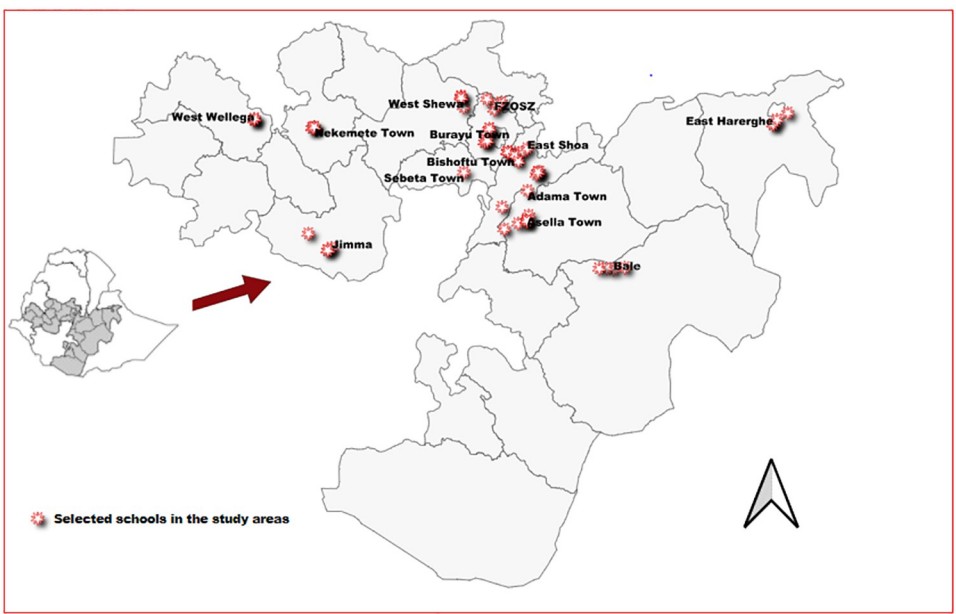

**Fig 1. Map showing the selected schools, zones, or towns for the seroprevalence of SARS CoV-2 study among students in Oromia, Ethiopia December 2020 to April 2021.**

The number of participants per district and school was assigned proportionally based on the number of students. The students were again proportionally allocated to public and private schools using Open Epi software: Toolkit Shell for Developing New Applications Population-based age-stratified seroepidemiological investigation protocol for coronavirus 2019 (COVID-19) infection. (who.int).

## Data collection and laboratory procedures

A standardized questionnaire was adopted from the national survey tool "*Serologic Evidence of SARS-CoV-2 Virus infection Survey in Ethiopia*, *June 2020*". It was used to collect demographic data, COVID-19-related symptoms, and prevention practices during the first week of school, reopening at the first-round baseline (December 2020), and, after four months, the final cohort (April 2021). The English version of the survey tools was translated into two widely spoken local languages (Afaan Oromo and Amharic). Two days of training were provided for the data collectors, sample collectors, and supervisors. The training included two sessions: survey tools and laboratory components. Questionaries were tested before use, and investigators supervised the supervisors and data collectors. The supervisors checked the completeness and consistency of the collected data using an online data entry software: CSentryCSProDataEntry7.2.1 https://www.census.gov/population/international/software/cspro/. The collected data was immediately sent online to the central server at Oromia Health Bureau for storage. GPS was taken for all schools, and spatial distribution of the schools was done by QGIS version 3.16.16 Hannover, 2009–2019 QGIS Development team. Ethiopian boundaries, districts, and towns shapefiles (https://data.humdata.org/dataset/cod-ab-eth) were used to construct the map (Fig 1).

Venous blood of about 5 mL was collected from all participants at each round using standard serum tubes. Any visible particulate matters in the specimen were removed by centrifugation at 3000 rounds per minute for 20 minutes at room temperature or by filtration. The sera were processed daily and stored at –20˚C in aliquots. One aliquot was subsequently thawed to

ensure the best reproducibility and cost-effective operation, and serology testing was done in batches. A serology test was performed with Elecsys anti-SARS-CoV-2 anti-nucleocapsid assay using Cobas 6000 module e601 system (Roche Diagnostics, Basel, Switzerland). The assay had a reported sensitivity of 100% and specificity of 99.81% in samples collected at least 14 days post-PCR confirmation of SARS-CoV-2 infection; 65.5% (95% CI 56.1–74.1) at 0–6 days and 88.1% (95% CI 77.1–95.1) at 7–13 days post-PCR confirmation [19]. The laboratory data consisted of qualitative data (reactive versus non-reactive) and quantitative data as a cut-off index (COI). Any COI value $<1$ was considered non-reactive, while $\geq 1$ was positive. Serology tests were conducted under the supervision of two professors from Jimma university. A double data entry process was done for sample results by two data managers, and no discrepancy was found.

## Data management and analysis

The data were exported from CSentry CSPro DataEntry 7.2.1 to Microsoft Excel to be cleaned and then exported to Stata version 14.2 for analysis. The result was summarized with frequencies and percentages. Seroprevalence at both rounds of study and seroincidence at the final cohort of the study was calculated, with point and interval estimates with 95% confidence intervals (CI). Stratified analyses were carried out to assess the prevalence and incidence of IgG in the different age subgroups, family size, mask utilization, physical distancing, and other variables. COVID-19 prevention practices were dichotomized as 'always' if they were implemented as the Ministry of Health recommended or 'not all' if they missed any component. Seroprevalence of anti-SARS-CoV-2 antibodies was calculated as the number of positive cases divided by the number of individuals tested per round. The incidence rate (IR) was calculated as the number of new positive cases divided by those still at risk of infection. The IR is presented as a rate per 100 000 person-weeks. Firstly, we conducted binary logistic regression for seroincidence, and multiple logistic regression was performed for variables with a p-value less than 0.2 in bivariate analysis. A p-value less than 0.05 was considered statistically significant.

## Ethical considerations

Ethical approval was received from the Oromia Health Bureau Review Board (Ref # BEFO/ HBTFH/1-16/234). In addition, a support letter to selected schools was obtained from the Oromia Education Bureau. Written informed consent was obtained from parents or caregivers. Assent was also obtained from children of age 12 years and older. All information accessed in the study was kept confidential through anonymity. Participants suspected of COVID-19 were referred to the nearest health facility for proper care. The findings of this study were officially communicated to health and education bureaus of the Oromia Region and presented to concerned stakeholders as a one-day workshop as soon as the results were available.

## Results

### Background characteristics of study participants

Although we aimed to include 2,164 school children overall, only 2024 (93.5%) gave blood samples at baseline. Among the 2024 collected samples, 140 were rejected due to sampling error and inadequacy. Finally, 1,884 were considered for seroprevalence calculation in the first round. Among 1,884 students approached for the second round, 613 were excluded from the sero incidence calculation for various reasons, leaving behind 1,271 participants for the seroincidence estimate (Fig 2).

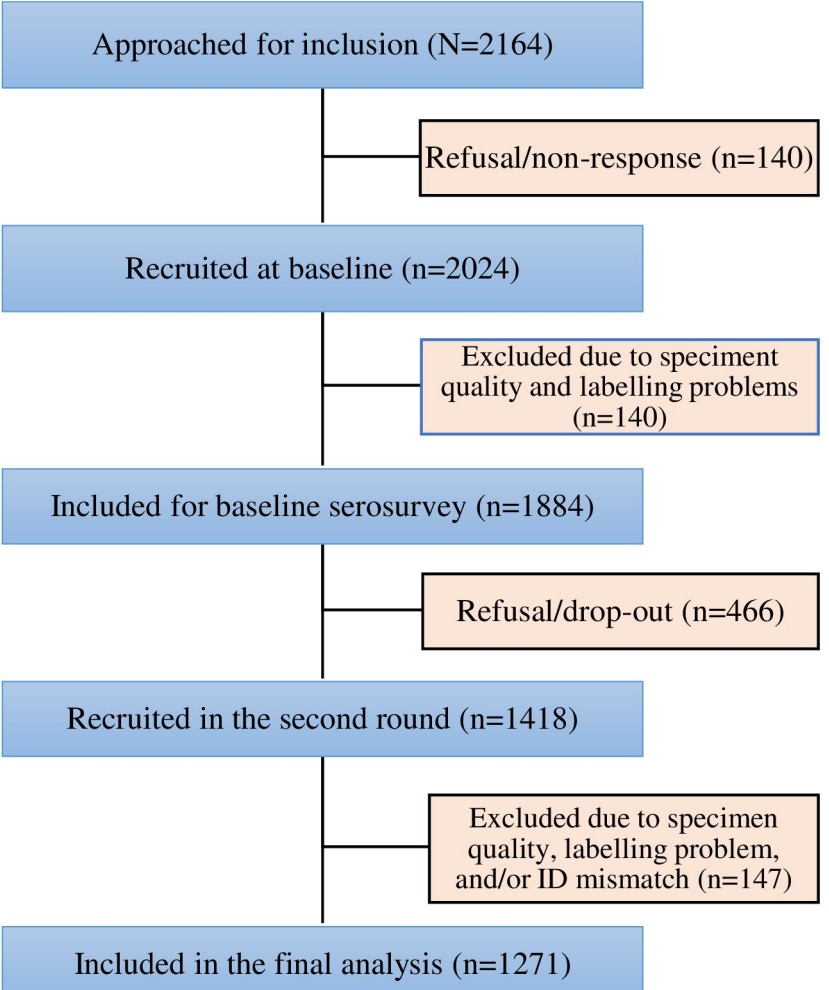

**Fig 2. Consort diagram representing cohort description for points of inclusion and exclusion (n = number of students at each step).** The mean (±SD) age of the participants at baseline was 15.8 (2.6) years, ranging from 10 to 27 years; 1116 (59.2%) of them were in the age group of 15 to 18 years. Females accounted for 1033 (54.8%) of the participants. The majority of the students, 1740 (92.4%), lived in a family size of three or more per household. Mask use and physical distance practices were low among the study participants. There are no marked differences in background characteristics between the first and second rounds of the students (Table 1).

### Seroprevalence and seroincidence

The overall seroprevalence of SARS-CoV-2 was 25.7% (485/1884) and 46.3% (588/1271) at the first round (baseline) and second-round (final cohort), respectively. Two-thirds of the school had a seroprevalence of more than 25% during the opening and doubled after four months of school reopening.

More than half (9/15) of the districts had a seroprevalence of > 25% at a school reopening. The prevalence doubled over four months in most districts except in Gimbi Town in west Oromia, which already had a seroprevalence of 42% during the first round. Overall highest prevalence during the two rounds was observed in districts surrounding the capital city, Addis Ababa (Fig 3).

However, the percent change in seroprevalence within the geographical area during the baseline study (December 2020) was inversely proportional to seroincidence during school

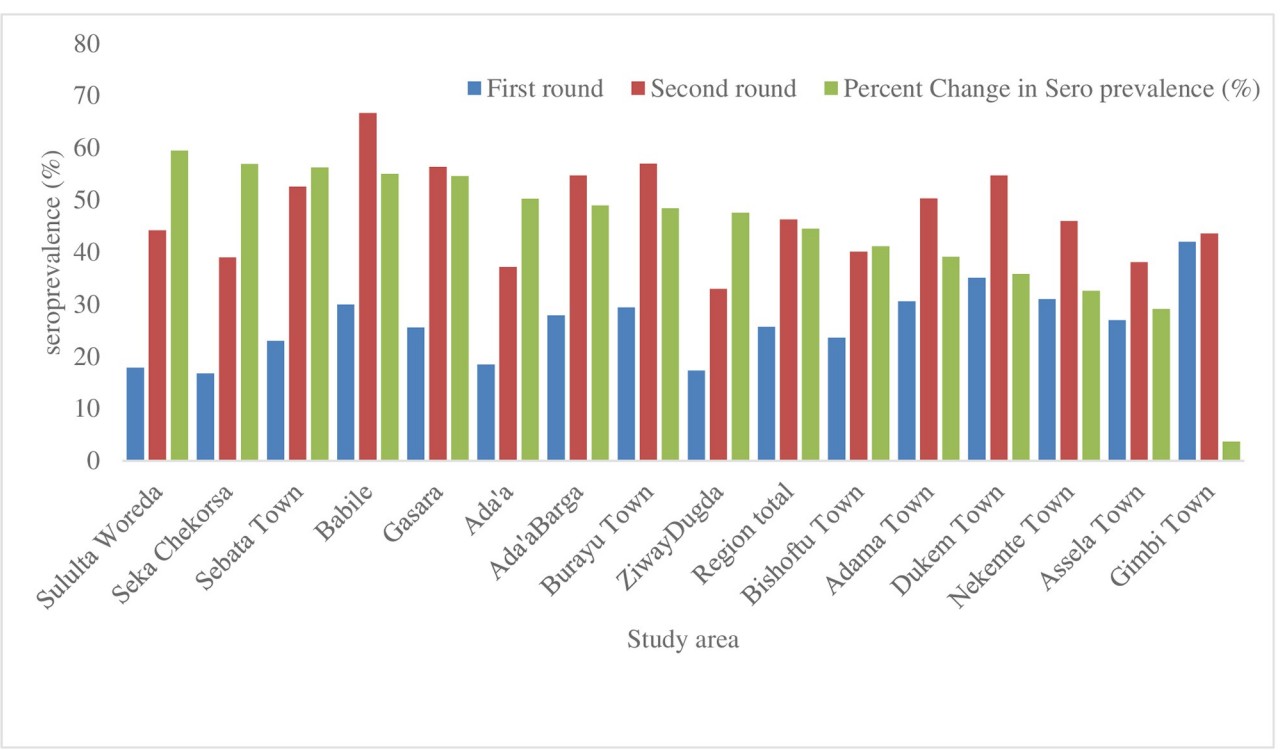

**Fig 3. Seroprevalence of SARS-CoV-2 and percent change in seroprevalence after school reopening in hotspot districts in Oromia, Ethiopia during study period, first round (December 2020) to second round (April, 2021).**

opening (April 2021) (Supporting information). Prevalence was higher in the age group 15–18 years and female students in both rounds (Table 2). There was a significant difference in seroprevalence among age group and school type. However, no significant difference in seroprevalence with mask utilization and physical distance practice was observed (Table 2).

The overall seroincidence during the study period was 1910 per 100,000 person-weeks. SARS-CoV-2 seropositivity was higher in high school students (grade nine to twelve) than in elementary school (grade four to eight) students ((RR = 1.2, 95%CI(1.21–2.22)). It was also higher in students with a family size of more than five((RR = 2.1, 95%CI(1.09–4.17)) than in fewer family sizes (Table 3).

## Availability of WASH and infection prevention practice materials and implementation status in schools (schools predictors for SARS-COV-2)

Among the surveyed 60 schools, only 23.3% had COVID-19 case definition and hotlines in the compound; 55% had posted COVID-19 prevention messages, 20% had a surveillance registration and reporting format, and only 58.3% had temporary isolation room (Supplementary Table 2). Water was not available for hand washing in 8.3% of the schools; 10% did not avail soap at hand washing sites. A physical distancing rule of at least 1 meter between people was implemented in only 5% of the schools (Supporting information). Hand washing facilities and sanitizers were always available for staff in 28.3% of schools. Adequate hand sanitizer for students was not available in 36.7% of the schools. On the other hand, sufficient no-touch dust bins with lids in all required places were always available in 13.3% of the schools (Supporting information).

**Table 1. Study district and participants' socio-demographic characteristics by study rounds, base line (December, 2020) and follow up (April, 2021).**

| Characterisitics | Categories | Baseline (N = 1884) n (%) | Follow-up (N = 1271) n (%) |
|---|---|---|---|
| **Study district** | Adea | 65 (3.5) | 43 (3.4) |
| | Adea Barga | 104 (5.5) | 53 (4.2) |
| | Adama Town | 360 (19.1) | 175 (13.8) |
| | Assela Town | 115 (6.1) | 84 (6.6) |
| | Babile | 50 (2.7) | 21 (1.7) |
| | Bishoftu Town | 165 (8.8) | 142 (11.2) |
| | Burayu Town | 136 (7.2) | 109 (8.6) |
| | Dukem Town | 57 (3.0) | 53 (4.2) |
| | Gasara | 78 (4.1) | 39 (3.1) |
| | Gimbi Town | 50 (2.7) | 39 (3.1) |
| | Nekemte Town | 113 (6.0) | 87 (6.9) |
| | Sebata Town | 200 (10.6) | 137 (10.8) |
| | Seka Chekorsa | 197 (10.5) | 146 (11.5) |
| | Sululta Woreda | 84 (4.5) | 43 (3.4) |
| | Ziway Dugda | 110 (5.8) | 100 (7.9) |
| **Age in Years** | 10–14 | 565 (30.0) | 402 (31.6) |
| | 15–18 | 1,116 (59.2) | 715 (56.3) |
| | >18 | 203 (10.8) | 154 (12.1) |
| **Sex** | Female | 1,033 (54.8) | 707 (55.6) |
| | Male | 851 (45.2) | 564 (44.4) |
| **Family size** | 1–2 | 144 (7.6) | 84 (6.6) |
| | 3–4 | 522 (27.7) | 357 (28.1) |
| | 5–7 | 938 (49.8) | 655 (51.5) |
| | > = 8 | 280 (14.9) | 175 (13.8) |
| **School-level** | Primary | 774 (41.1) | 540 (42.5) |
| | Secondary | 1110 (58.9) | 731 (57.5) |
| **Mask utilization** | Always | 502(26.7) | 367 (28.9) |
| | Never | 243 (12.9) | 124 (9.8) |
| | Occasionally | 1139 (60.5) | 780 (61.4) |
| **Physical distancing practice** | Always | 149 (7.9) | 98 (7.7) |
| | Never | 654 (34.7) | 495 (39.0) |
| | Occasionally | 1081 (57.4) | 678 (53.3) |
| **Contact history with** | Suspected COVID-19 | Yes | 15 (0.8) | 15 (1.2) |
| | Confirmed COVID-19 | Yes | 12 (0.6) | 13 (1.0) |
| **Self-reported symptoms** | Fever | Yes | 73 (3.9) | 39 (3.1) |
| | Cough | Yes | 199 (10.6) | 91 (7.3) |
| | Difficult of breathing | Yes | 5 (0.3) | 8 (0.6) |
| | Chest pain | Yes | 6 (0.3) | 13 (1.0) |
| | Wheezing | Yes | 26 (1.4) | 45 (3.6) |
| | Fatigue | Yes | 26 (1.4) | 30 (2.4) |

Frequently touched objects were properly cleaned and disinfected with 0.5% chlorine solution in only 11.7% of the schools. Screening for COVID_19 was appropriately practiced in only 16.7% of the schools. Ways of waste collection, handling, and disposal methods were safe in 28.3% of the schools. In contrast, 35% of schools didn't look at all practice identification of sick students (Supporting information).

**Table 2. Seroprevalence of SARS -CoV-2 (seroprevalence = n/N*100, N = Total participants and n = SARS-COV-2 positive for IgG, Where N = 1884 and n = 485 for the first round and N = 1721 and n = 588 for second round respectively).**

| Charectercitcs | Category | First round (December, 2020) | | Second round (April, 2021) | |
|---|---|---|---|---|---|
| | | Seroprevalence(%) | P-value | Seroprevalence(%) | P-value |
| **Age in Years** | 10_14 | 124 (6.6%) | ref | 155 (12.2%) | ref |
| | 15–18 | 293 (15.5%) | 0.054 | 351 (27.6%) | 0.001 |
| | >18 | 68 (3.6%) | 0.001 | 82 (6.5%) | 0.002 |
| **Sex** | Female | 261 (13.8%) | ref | 320 (25.2%) | ref |
| | Male | 224 (11.9%) | 0.520 | 268 (21.1%) | 0.423 |
| **Family size** | 1_2 | 33 (1.8%) | ref | 32 (2.5%) | ref |
| | 3_4 | 136 (7.2%) | 0.444 | 163 (12.8%) | 0.21 |
| | 5_7 | 252 (13.4%) | 0.317 | 310 (24.4%) | 0.112 |
| | > = 8 | 64 (3.4%) | 0.989 | 83 (6.5%) | 0.158 |
| **School** | Primary school | 170 (9.0%) | ref | 219 (17.2%) | ref |
| | Secondary school | 315 (16.7%) | 0.002 | 369 (29.1%) | 0.0001 |
| **Mask Utilization** | Always | 141 (7.5%) | ref | 182 (14.3%) | ref |
| | Never | 52 (2.8%) | 0.051 | 52 (4.1%) | 0.141 |
| | Occasionally | 292 (15.5%) | 0.299 | 354 (27.9%) | 0.183 |
| **Physical Distance practicing** | Always | 48 (2.5%) | ref | 47 (3.7%) | ref |
| | Never/Rarely | 167 (8.9%) | 0.236 | 214 (16.8%) | 0.390 |
| | Occasionally | 270 (14.3%) | 0.156 | 327 (25.7%) | 0.960 |
| **Having an underlying medical condition** | No | 458 (24.3%) | ref | 580 (45.6%) | ref |
| | Do not know | 4 (0.2%) | 0.384 | 3 (0.2%) | 0.845 |
| | Yes | 23 (1.2%) | 0.021 | 5 (0.4%) | 0.123 |
| **Contact history with suspected COVID_19** | No | 481 (25.5%) | ref | 578 (45.5%) | ref |
| | Yes | 4 (0.2%) | 0.935 | 10 (0.8%) | 0.121 |

## Discussion

The seroprevalence of SARS-CoV-2 almost doubled within the four months of schools reopening among students in COVID-19 hotspot districts of the Oromia Region. The seroincidence was high among high school students (> = grade 9), those living in extended family size. SARS-COV-2 seroincidence was higher in high school students (grades 9 to 12) than elementary school (grades four to eight) students and among students with large family sizes (> 5) than those with smaller family sizes (<3). This seroepidemiological change coincided with the second wave of the COVID-19 outbreak in Ethiopia, characterized by a health system crisis during the first quarter of 2021 [3], indicating the possible contribution of school opening to the new wave of the outbreak.

Besides its fatality and health system crisis, the COVID-19 pandemic has caused extreme psychosocial crises globally due to the implementation of stringent public health measures by almost all countries [20]. Although children have never faced the pandemic, they have become the most prominent victims [21]. Due to their potential to perpetuate infection spread, schools were closed early to contain the pandemic confining students to their homes away from schools and their peers with looming uncertainties. In Ethiopia alone, over 26 million children were kept away from school for nearly nine months [22]. Such lockdown measure was challenging for children, families, and the country. Despite their proven effectiveness, lockdown measures lead to general fatigue and poor adherence to mitigation measures the longer they are kept in place.

**Table 3. Multiple logistic regression of SARS-CoV-2 and association with seroincidence, December 2020 to April 2021.**

| Characterictics | Categories | Sero incidence (N = 918) | | RR | (95%, Conf. Interval) | | P-Value |
|---|---|---|---|---|---|---|---|
| | | IgG Negative for SARS-CoV-2 (n = 636) | IgG Positive for SARS-CoV-2 (n = 282) | | Lower | Upper | |
| School | Primary | 300 (32.7%) | 107 (11.7%) | ref | | | |
| | Secondary | 336 (36.6%) | 175 (19.0%) | 1.46 | 1.21 | 2.22 | 0.002 |
| Mask Utilization | Frequently | 174 (19.0%) | 73 (8.0%) | Ref | | | |
| | Occasionally | 398 (43.4%) | 180 (19.6%) | 0.96 | 0.66 | 1.41 | 0.842 |
| | Rarely | 64 (7.0%) | 29 (3.2%) | 1.02 | 0.56 | 1.84 | 0.958 |
| Sex | Female | 362 (39.4%) | 149 (16.2%) | ref | | | |
| | Male | 274 (29.8%) | 133 (14.5%) | 1.21 | 0.90 | 1.63 | 0.202 |
| Family Size | 1–2 | 50 (5.4%) | 12(1.3%) | ref | | | |
| | 3–4 | 178 (19.4%) | 76 (8.3%) | 2.00 | 1.00 | 4.01 | 0.051 |
| | 5–7 | 323 (35.5%) | 152 (16.6%) | 2.13 | 1.09 | 4.17 | 0.027 |
| | > = 8 | 85 (9.3%) | 42 (4.6%) | 2.27 | 1.07 | 4.79 | 0.032 |
| Presence BCG Scar | Yes | 470 (51.2%) | 204 (22.2%) | ref | | | |
| | No | 166 (18.1%) | 78 (8.5%) | 1.08 | 0.78 | 1.49 | 0.654 |
| Attending or participating in different events | Frequently | 30 (3.3%) | 10 (1.1%) | ref | | | |
| | Never | 440 (47.9%) | 199 (21.7%) | 1.44 | 0.67 | 3.08 | 0.351 |
| | Sometimes | 166 (18.1%) | 73 (8.0%) | 1.30 | 0.58 | 2.89 | 0.522 |
| participating in different worship activities | Frequently | 66 (7.2%) | 24 (2.6%) | Ref | | | |
| | Never | 321 (35.0%) | 141 (15.4%) | 1.16 | 0.67 | 2.01 | 0.607 |
| | Sometimes | 249 (27.1%) | 117 (12.7%) | 1.24 | 0.72 | 2.14 | 0.446 |
| Using public transport | Frequently | 119 (13.0%) | 43 (4.7%) | Ref | | | |
| | Never | 306 (33.3%) | 134 (14.6%) | 1.32 | 0.84 | 2.06 | 0.231 |
| | Sometimes | 211(23.0%) | 105 (11.4%) | 1.39 | 0.87 | 2.22 | 0.163 |
| Practicing physical distancing | Frequently | 47(5.1%) | 15 (1.6%) | Ref | | | |
| | Never | 327(35.6%) | 158 (17.2%) | 1.52 | 0.78 | 2.98 | 0.218 |
| | Sometimes | 262(28.5%) | 109 (29.4%) | 1.28 | 0.63 | 2.58 | 0.497 |

Ethiopia appeared to have controlled the outbreak in the first six months. Yet, surveillance data revealed that the infection spread steadily during these times [23]. The 'silent' spread might have gotten momentum once most mitigation measures were relaxed in September 2020. This is probably why seroprevalence was high (25.7%) among children even before school commencement. School reopening when sustained disease spread inevitably fuels its propagation [24, 25]. For instance, the RT-PCR test positive rate in Ethiopia, which was 8.5% on average for December 2020 and January 2021, increased to 12.5% in February 2021, 21.1% in March, and 22.5% for April 2021 (Figs 3 and 4).

Although multiple other factors might have contributed, it is thus fair to assume that the high SARS-CoV-2 seroincidence with prevalence reaching 46.3% by April 2021 among students in our study might have played a critical role in the new wave of the outbreak. Seroprevalence varied within the geographical areas. During the baseline study (December 2020), seroprevalence was highest around the capital city (Addis Ababa) except in Gimbi Town. The percentage change in seroprevalence within the geographical area is inversely related to an incidence, as indicated in (Supporting information). The change in seroprevalence from the town around the capital to the peripheral district in the region during the school reopening indicates high community transmission.

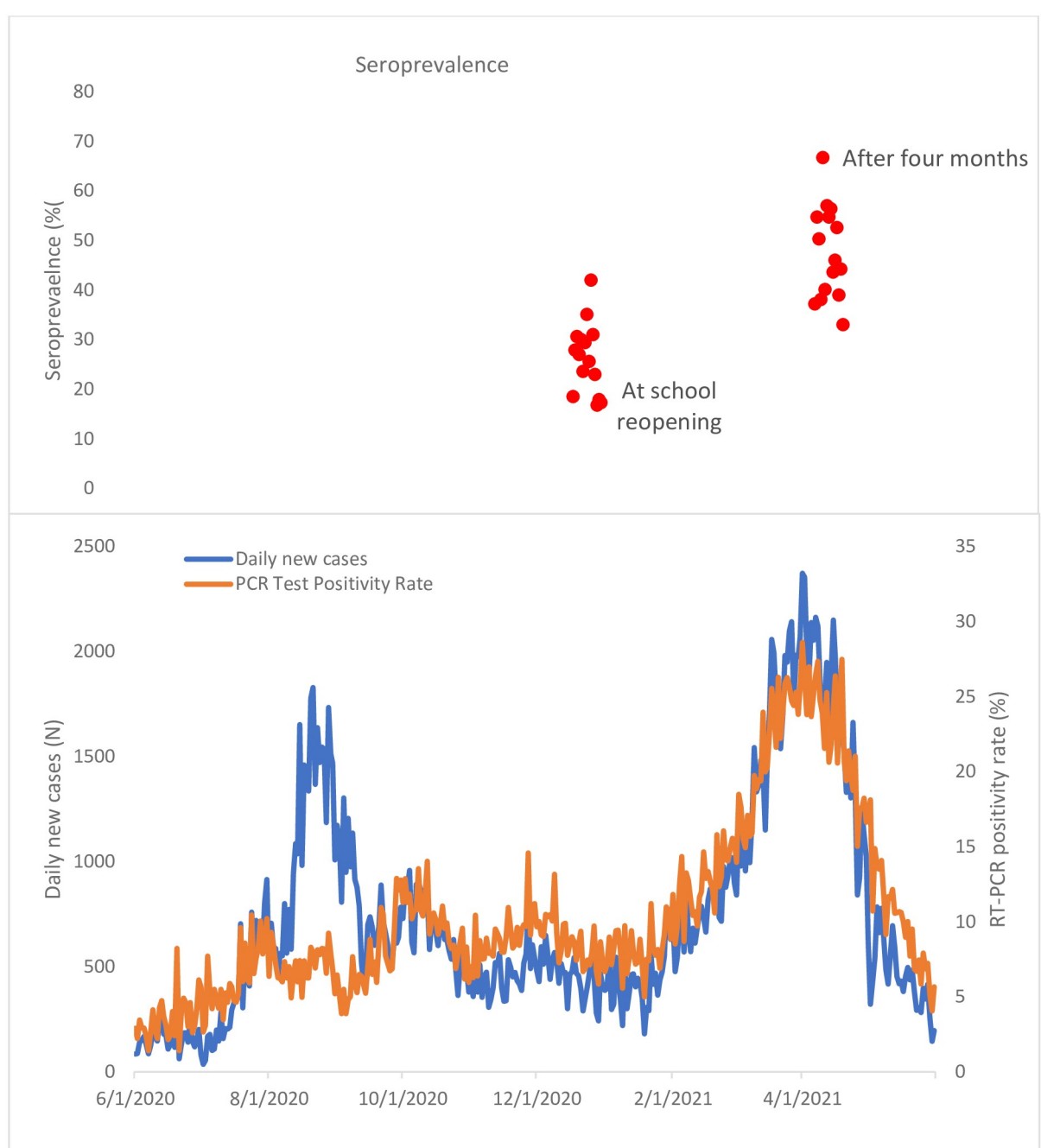

**Fig 4. Trend changes in SARS -CoV-2 seroprevalence in hotspot districts, national RT-PCR positivity rate and daily new cases.** Each orange dot represents hotspot districts. Data from the national RT-PCR positivity rate and daily new cases were obtained from the daily COVID-19 report by the Ministry of Health of Ethiopia [3].

This study finding is also congruent with the seroepidemiological survey of SARS-CoV-2 among hospital workers and communities in Ethiopia that revealed an escalation of seroprevalence from 11% in August 2020 to 54% in April 2021 [26].

The risk of infection in younger children has been lower than in adults. However, the incidence of the infection increases with age [27, 28]. In our study, the seroincidence was higher

among high schools than in primary school, which may be related to age. Although there is no well-established evidence that children are less susceptible to SARS-CoV-2 infection, the infection tends to be milder and predominantly asymptomatic [29, 30] in this age group leading to under-reporting of the case. Besides, as the social interaction in younger children is limited, their risk of getting and transmitting the expected infection is lower than that of adolescents and adults [27, 31], as has been the case in our report and previous studies.

Our study's seroincidence rate peaked at 1,910 per 100,000 person-weeks in April 2021. It is far higher than the national routine surveillance report from December 2020 to April 2021, although the incremental trend was the same in both cases (Fig 4). This incidence rate was lower than that of health workers in February 2021 (IR 2223 per 100,000 person-weeks in Addis Ababa and 3810 per 100,000 person-week in Jimma) [26]. The findings in these two studies underline the serious underreporting of COVID-19 cases in Ethiopia in general and in children in particular. Our study suggests that seroprevalence change coincided with a change in Ethiopia's officially reported SARS-CoV-2 RT-PCR positivity rate after school reopening. The escalation of seroincidence and dramatic increment of PCR positivity suggests that school opening might have contributed to the COVID-19 peaks that occurred in the country in March/April 2021 (Fig 4).

A high refusal rate was reported in big towns and private schools (Table 1). It was expected that a high non-response rate can be reported in a follow-up study, and it is possible to estimate both seroepidemiology and seroincidence within the given sample size [32–36]

COVID-19 prevention strategies at all surveyed schools were far below the target set nationally to be implemented in all schools, both in high schools and elementary as well as in private and government schools (School Reopening Guidance in the Context of COVID-19 Pandemic in Ethiopia). The guideline states that the local task force (authority) should decide whether to reopen schools or not based on the local pattern of COVID-19 outbreak and schools' capability to implement COVID-19 prevention measures [55]

It also states that there should be a sufficient classroom with a limited number of students to keep a one-meter distance between desks, only one student per desk, and a 4-meter distance between teachers' stand to front students. The requirements were adequate and continuous water supply, sufficient laterin, and sufficient washing facilities [55]

These findings indicate that most schools were not ready to implement the recommended measures at the time of school reopening and could not adhere to them after school commencement. It is thus highly likely that the poor adherence of the schools and students to these measures is the main reason for the high SARS-CoV-2 seroincedence after school reopening.

The high seroincidence and doubling seroprevalence in children during school reopening in this study implies: that the Ethiopian education sector, which encompasses nearly 30% of the general population (as stated in the Supporting information), suggests as school is a potential source of infection for COVID-19 and other emerging and re-emerging epidemic-prone disease which needs strong policy, strategies, and planning at school while implementing any pandemic /Epidemic control and preventive measures.

## Strength and limitation

The major strengths of this study were that it covered a wide geographical area and was a longitudinal follow-up study. Moreover, the Elecsys anti-SARS-CoV-2 test used for this study is internationally acceptable and standard for estimating the seroprevalence and seroincidence. Based on our information, this is the only longitudinal study conducted on children and schools in the country. Perhaps, there were some limitations, like the high dropout rate of study participants in big towns and some other rural areas because of fear of giving blood

samples, insecurity, and fasting time for the Muslim community during the study follow-up period. Furthermore, we conducted the seroprevalence survey only twice and four months apart. As a result, we could not see the seroepidemiological pattern more accurately. Our study is also limited because we only conducted it in hotspot districts, most of which were urban or semi-urban areas. Hence, our findings may not reflect the region's real outbreak pattern.

## Conclusion

SARS-CoV-2 seroprevalence among students aged 10 years and above in the Oromia Region hotspot zones was high even before school reopening and almost doubled at the four-month post-school opening. Seroincidence was higher in high school students (grades 9–12) and children with a family size of more than five. The nationally recommended COVID-19 prevention measures were not adequately implemented in most schools. The findings reflect the challenges of implementing social restrictions and other public health measures in low-income settings as part of pandemic control measures. The rapid escalation of the seroprevalence happened during the second wave of the COVID-19 outbreak in Ethiopia, indicating a possible contribution of school reopening for the new outbreak wave. Hence, any pandemic/outbreak control measures should include school health as a critical component of public health measures.

## Supporting information

**S1 File. Supporting information consiting of all supporting tables and figures.**
(DOCX)

## Acknowledgments

We greatly appreciate the collaborative efforts from Oromia Health Bureau, Fenot Project, Jimma University, Armauer Hansen Research Institute, and Adama Reginal Laboratory and staff for the overall coordination of the study. Their contribution from the concept note development, during data and sample collection, providing technical support, offering test kits, sample storage, sample transportation, data analysis, and report writing. We would also like to thank the local administrators of the study zones and towns, data collectors, school principals, and teachers. We truly thank all children and their parents for their willingness to participate in this study.

## Author Contributions

**Conceptualization:** Dabesa Gobena, Esayas Kebede Gudina, Daniel Yilma, Tsinuel Girma, Getu Gebre, Tesfaye Gelanew, Alemseged Abdissa, Daba Mulleta, Tarekegn Sarbessa, Henok Asefa, Mirkuzie Woldie, Gemechu Shumi, Birhanu Kenate, Arne Kroidl, Andreas Wieser, Beza Eshetu, Tizta Tilahun Degfie, Zeleke Mekonnen.

**Data curation:** Dabesa Gobena, Esayas Kebede Gudina, Daniel Yilma, Tsinuel Girma, Getu Gebre, Tizta Tilahun Degfie, Zeleke Mekonnen.

**Formal analysis:** Dabesa Gobena, Esayas Kebede Gudina, Tesfaye Gelanew, Henok Asefa, Tizta Tilahun Degfie, Zeleke Mekonnen.

**Funding acquisition:** Dabesa Gobena, Esayas Kebede Gudina, Gemechu Shumi, Birhanu Kenate, Tizta Tilahun Degfie, Zeleke Mekonnen.

**Investigation:** Dabesa Gobena, Esayas Kebede Gudina, Tizta Tilahun Degfie, Zeleke Mekonnen.

**Methodology:** Dabesa Gobena, Esayas Kebede Gudina, Daniel Yilma, Tsinuel Girma, Tesfaye Gelanew, Alemseged Abdissa, Daba Mulleta, Henok Asefa, Mirkuzie Woldie, Tizta Tilahun Degfie, Zeleke Mekonnen.

**Project administration:** Dabesa Gobena, Tizta Tilahun Degfie, Zeleke Mekonnen.

**Resources:** Dabesa Gobena, Gemechu Shumi, Birhanu Kenate, Tizta Tilahun Degfie, Zeleke Mekonnen.

**Software:** Dabesa Gobena, Henok Asefa, Tizta Tilahun Degfie, Zeleke Mekonnen.

**Supervision:** Dabesa Gobena, Esayas Kebede Gudina, Daniel Yilma, Tsinuel Girma, Getu Gebre, Tesfaye Gelanew, Alemseged Abdissa, Daba Mulleta, Tarekegn Sarbessa, Henok Asefa, Mirkuzie Woldie, Gemechu Shumi, Birhanu Kenate, Arne Kroidl, Andreas Wieser, Beza Eshetu, Tizta Tilahun Degfie, Zeleke Mekonnen.

**Validation:** Dabesa Gobena, Esayas Kebede Gudina, Tesfaye Gelanew, Alemseged Abdissa, Daba Mulleta, Tarekegn Sarbessa, Henok Asefa, Mirkuzie Woldie, Gemechu Shumi, Birhanu Kenate, Arne Kroidl, Andreas Wieser, Beza Eshetu, Tizta Tilahun Degfie, Zeleke Mekonnen.

**Visualization:** Dabesa Gobena, Esayas Kebede Gudina, Daniel Yilma, Tsinuel Girma, Getu Gebre, Tesfaye Gelanew, Alemseged Abdissa, Daba Mulleta, Henok Asefa, Mirkuzie Woldie, Gemechu Shumi, Birhanu Kenate, Arne Kroidl, Andreas Wieser, Beza Eshetu, Tizta Tilahun Degfie, Zeleke Mekonnen.

**Writing – original draft:** Dabesa Gobena, Tizta Tilahun Degfie, Zeleke Mekonnen.

**Writing – review & editing:** Dabesa Gobena, Esayas Kebede Gudina, Daniel Yilma, Tsinuel Girma, Getu Gebre, Tesfaye Gelanew, Alemseged Abdissa, Daba Mulleta, Tarekegn Sarbessa, Henok Asefa, Mirkuzie Woldie, Gemechu Shumi, Birhanu Kenate, Arne Kroidl, Andreas Wieser, Beza Eshetu, Tizta Tilahun Degfie, Zeleke Mekonnen.

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
