## [Decision Letter · Decision Letter 0]

19 Jul 2022

PONE-D-22-12485Escalating spread of SARS-CoV-2 infection after school reopening among students in hotspot districts of Oromia Region in Ethiopia: Longitudinal studyPLOS ONE

Dear Dr. Gobena,

Thank you for submitting your manuscript to PLOS ONE. After careful consideration, we feel that it has merit but does not fully meet PLOS ONE’s publication criteria as it currently stands. Therefore, we invite you to submit a revised version of the manuscript that addresses the points raised during the review process.=

We look forward to receiving your revised manuscript.

Kind regards,

Md. Tanvir Hossain

Academic Editor

PLOS ONE

Journal Requirements:

2. We note that Figure 1 in your submission contain Map images which may be copyrighted. All PLOS content is published under the Creative Commons Attribution License (CC BY 4.0), which means that the manuscript, images, and Supporting Information files will be freely available online, and any third party is permitted to access, download, copy, distribute, and use these materials in any way, even commercially, with proper attribution. For these reasons, we cannot publish previously copyrighted maps or satellite images created using proprietary data, such as Google software (Google Maps, Street View, and Earth). For more information, see our copyright guidelines: http://journals.plos.org/plosone/s/licenses-and-copyright.

Natural Earth (public domain): http://www.naturalearthdata.com

Reviewers' comments:

Reviewer's Responses to Questions

**Comments to the Author**

1. Is the manuscript technically sound, and do the data support the conclusions?

Reviewer #1: Yes

Reviewer #2: Partly

2. Has the statistical analysis been performed appropriately and rigorously? 

Reviewer #1: Yes

Reviewer #2: No

3. Have the authors made all data underlying the findings in their manuscript fully available?

Reviewer #1: No

Reviewer #2: Yes

4. Is the manuscript presented in an intelligible fashion and written in standard English?

Reviewer #1: Yes

Reviewer #2: Yes

5. Review Comments to the Author

Reviewer #1: Thank you for the interesting article. I have few observations that author should consider:

1. Please clarify the selection of participants, specifically the last paragraph, "the margin of error of 4%, 95% confidence interval". The sampling and sampling size calculation needs more clarification with references.

2. "Finally, the sample size computed was 2164 students". However, in Results section "We aimed to include 2164 students; however, we could only recruit 1884 in the first round and retained only 1271 for the second round. The rest were excluded for various reasons". Then how can you ensure your calculated sample size?

3. How can the authors maintain the validity of the questionnaire for these school children?

4. Please clarify the data analysis part, mention all the analysis that authors performed.

5. I suggest to elaborate the results part, it looks too short.

Reviewer #2: Evidence before this study

1. Please add reference in this section.

Implications of all of the available evidence

2. “Although it is possible that this outbreak pattern change could have also been due to the arrival of the new variants of the virus” please add reference.

Introduction:

You have mention “This study, therefore, aimed to report on the seroepidemiological changes and predictors among students during school closure and after reopening in hotspot districts in

Oromia Region, Ethiopia.”. Have find any predictors? Please add those in the results section with interpretation and discussion section with justification.

Methods:

Selection of study participants

3. “Four schools were chosen from each district, making 60 schools overall”. How many schools were there for different district? Why four school were chosen? Was that representative to others?

4. You have utilized Multiple Logistic Regression of SARS-CoV-2 and association with seroincidence, please mention the data creating process and model fitting information. This model shows that two variables are associated. Please interpret the results of this table.

5. As your study is a longitudinal study, have you used any longitudinal analysis method for analyzing your data?

Discussion:

6. Please try to relate your discussion with your findings.

6. PLOS authors have the option to publish the peer review history of their article (what does this mean?). If published, this will include your full peer review and any attached files.

Reviewer #1: No

Reviewer #2: No

---

## [Author Response · Author response to Decision Letter 0]

3 Oct 2022

Point-by-point responses to reviewer’s and editor's comments 

Journal: PLoS One

MS ID: PONE-D-22-12485

Dear Editor in Chief, 

We thank you and the reviewers for your appraisal and constructive comments on our manuscript titled 'Escalating spread of SARS-CoV-2 infection after school reopening among students in hotspot districts of Oromia Region in Ethiopia: Longitudinal study". We have provided a detailed response to each reviewer’s comment. In addition, the clean (unmarked) and marked-up versions of the manuscript were highlighted in yellow to show changes made. 

We look forward to hearing your positive feedback and consideration of the revised manuscript for publication. 

Sincerely,

Dabesa Gobena, Corresponding Author

Reviewer # 1

 Please clarify the selection of participants, specifically in the last paragraph, "the margin of error of 4%, 95% confidence interval". The sampling and sampling size calculation needs more clarification with references.

Response: Thank you for your comment, and here below are the clarification points. 

Initially, the top 15 Covid-19 hotspot areas (7 towns and 8 woredas) in the Oromia region were purposively taken (Supplementary table 1). The schools in each town and woreda were identified and classified into primary schools (public and private) and high schools (public and private). One school from the above four categories was randomly selected and stratified by grades (4 to 12), and then one class from each grade was chosen randomly. It was automatically considered if there was only one class in each grade. Finally, students were randomly selected from each selected class using a computer-based lottery method. The school types and grades were stratified to obtain a good representation (age 10+ years) from all grades (see Supplementary Figure 1).

The primary objective of the study “determining the proportion of children who generated an antibody response during school closure was used to calculate sample size and allocation”. A one-population proportion formula was used to determine the required sample size. The assumptions taken to compute the sample size were: the proportion generated antibody response during school closure to be 50%, 4% of margin of error, and 95% confidence interval, and the design effect of (de = 3), since there were three selections (schools from each hot spot areas, class from each grade and students from each class), 10% non-response rate and 10% dropouts, by using the following formula, the final sample size was 2,164 school-age children. 

The required sample size was: n=(〖Z_(α⁄2)〗^2 P(1-P))/ϵ^2 = (〖1.96〗^2 X0.5(1-0.5))/〖0.04〗^2 =601*3*0.1*0.1≅2164

Finally, the total sample size was assigned proportionally to the towns and woredas based on the number of students. The students were again proportionally allocated to public and private schools using Open Epi software. (Open Epi - Toolkit Shell for Developing New Applications Population-based age-stratified seroepidemiological investigation protocol for coronavirus 2019 (COVID-19) infection (who. int). 

 "Finally, the sample size computed was 2164 students". However, in the Results section, "We aimed to include 2164 students; however, we could only recruit 1884 in the first round and retained only 1271 for the second round. The rest were excluded for various reasons". Then how can you ensure your calculated sample size?

Response: Thank you for your comments and below are the clarification points. 

Of the 2,164 sample size computed to participate in the prospective cohort of the study, 2024 (93.5%) children provided sa ample, whereas 140 (6.5%) refused to give a sample. Among the 2024 collected samples, 140 were rejected due to sampling error and sample inadequacy. Lastly, 1,884 were considered for seroprevalence calculation in the first round. Among 1,884 students approached for the second round, 613 were not included for sero-incidence calculation due to refusal, sampling error, and sample inadequacy during the first round. Thus, 1,271 school-age children with complete data were used to calculate sero-incidence (Figure 2- in the manuscript). An elevated refusal rate was reported in big towns and private schools. 

Moreover, in a follow-up study, it is usual to experience a high non-response rate which was the case in the present study. Nevertheless, we feel that the current sample size (1271) suffices to estimate both seroepidemiology and sero-incidence. We have included supporting references (32,33,34,35,36), ensuring our calculated sample size is acceptable.

 How can the authors maintain the validity of the questionnaire for these school children?

Response: Thank you for your comment and below are the clarification points. 

We used standardized survey questionnaires adapted from the WHO protocol for serosurveys, Africa Centers for Disease Control and Prevention, and nationally prepared questionnaire serosurvey, and school preparedness checklists were used to collect data using tablets. We have used CSentryCSProDataEntry7.2.1 software for data capturing. The English version of the survey tools was translated into two local languages, Afaan Oromo and Amharic, since both languages are used as the mother tongue of students in the selected schools. Two days of training were given to the data collectors, sample collectors, and supervisors. The training includes two sessions: survey tools and laboratory components. The supervisors check the data collectors using an online CSentryCSProDataEntry7.2.1 https://www.census.gov/population/international/software/cspro/ application software about the survey tools' completeness and data quality. The questionnaire was piloted in 5% of the schools in Adama rural district before use, and the research team supervised the supervisors and data collectors. When data were entered into the data collection software, it was accessible online to the central server, and it was possible to import STATA version 14.2 for data management and analysis. GPS data coordinates were taken for all schools. Furthermore, a double data entry process was done for sample results by two data managers and no discrepancies between the different data managers

 Please clarify the data analysis part, mention all the analyses that the authors performed.

Response: Thank you for your feedback. Below are the clarification points. 

The data were exported from CSentryCSProDataEntry7.2.1 application software. Data management was done using Excel, while Stata version 14.2 was used for data analysis. Results were summarized with absolute and relative (percentage) frequencies. Seroprevalence at both rounds of study and seroincidence at the final cohort of the study was calculated, with point and interval (95% confidence intervals, CI) estimates. Stratified analyses were conducted to assess the prevalence of IgG in the different age subgroups, family size, mask utilization, Physical distancing, and other variables. We performed binary logistic regression for seroincidence and multiple logistic regression for variables with a p-value less than 0.25. A p-value less than 0.05 was considered statistically significant. All statistical computations were performed with the statistical software Stata version 14.2. 

 I suggest elaborating the results part; it looks too short.

Response: Thank you for your comment. We accepted the comment, and revisions were made to the result part in the revised manuscript. Most of the result parts were presented in figures and tables

Reviewer #2

Evidence before this study

1. Please add references in this section.

Implications of all of the available evidence

Response: Dear Reviewer, thank you for your comment. 

We included the available evidence in the manuscript before and during our study period, such as Ref# (37 to 48).

2. "Although it is possible that this outbreak pattern change could have also been due to the arrival of the new variants of the virus" please add reference.

Response: Thank you for your comment. We accepted the comment and included the reference in the main manuscript. 

"The emergence and rapid global spread of the new Delta and, more recently, Omicron variants of SARS-CoV-2 pose a daunting public health emergency. As an RNA virus, the covid-19 virus continues to mutate, resulting in the emergence of new variants with high transmissibility, such as the recently discovered Omicron variant. Ref# (49 to 54)."

Introduction:

You have mentioned "This study, therefore, aimed to report on the seroepidemiological changes and predictors among students during school closure and after reopening in hotspot districts in

Oromia Region, Ethiopia.". Have find any predictors? Please add those in the results section with interpretation and discussion section with justification.

Response: Thank you for your comment. 

We incorporated findings from schools in the results section. It is about the predictors of seroepidemiological change, and a supplementary table is attached. The justification in both the result and discussion is highlighted in yellow. 

Methods:

Selection of study participants

3. "Four schools were chosen from each district, making 60 schools overall". How many schools were there for different district? Why four school were chosen? Was that representative to others?

Response: Thank you for your comment; here are some clarifications. 

First, it seems there is a misunderstanding. It is not meant four schools were chosen from each district, rather, four types of schools (private primary, public primary, private high schools, and public high schools). Overall, 60 schools were chosen, and we feel those 60 schools would represent others. For further clarification, briefly, the selection was made as follows:

Primarily, the top 15 Covid-19 hotspot areas (7 towns and 8 woredas) in the Oromia region were purposively considered in this study (Supplementary table 1). A total of 1,032 (both elementary and high schools) were found in the selected hotspot districts. Then, schools in each town and woreda were identified and classified into four categories: primary schools (public and private) and high schools (public and private). One school from each of the above four categories was randomly selected and stratified by grades (4 to 12), and then one class from each grade was chosen randomly. It was automatically considered if there was only one class in each grade. Finally, students were randomly selected from each selected class using a computer-based lottery method. The school types and grades were stratified to get obtain a good representation (age 10+ years) from all grades. Regarding the representativeness of the schools, a single confirmed SARS-COV-2 can be considered the region's epidemic, potentially transmitting the disease to the wider community, and students from different residential areas within the districts were randomly selected from schools. 

4. You have utilized Multiple Logistic Regression of SARS-CoV-2 and association with seroincidence, please mention the data creating process and model fitting information. This model shows that two variables are associated. Please interpret the results of this table.

Response: Thank you for your comment and time to review our work.

We accepted the comment, and interpretations of the tables were incorporated into the discussion. Concerning the data-creating process and model fitness, all data cleaning and analysis were done by STATA version 14.2. First, we generated the incidence by removing the positive cases during the first round, replacing, labeling value, label defines, and desiring to change string variable to numeric was conducted for each variable accordingly. Binary Logistic regression for each independent variable with incidence (Outcome variable) was performed. Independent variables with P-Value <= 0.2 were transferred to a multiple logistic regression model, and variables with P-Value less than 0.05 were considered as SARS -COV-2 sero incidence change predictors. Relative risk was calculated to measure the strength of the association. The likelihood of PseduoR2 between zero and one was used to check the Model fitness. 

5. As your study is a longitudinal study, have you used any longitudinal analysis method for analyzing your data?

Response: Thank you for your feedback.

We have used longitudinal data analysis methods to look for changes in SARS-COV-2 seroprevalence at both rounds, seroincidence, and predictors at the second round among school children. 

Discussion:

6. Please try to relate your discussion with your findings.

Response: Thank you for your feedback.

We have incorporated your feedback into the discussion and results. The discussion part is revised by considering the sections of the results part. 

Response to Editor, Copyright issues of figure 1 

Response: Dear Editor, thank you again for your time and comments in reviewing our works. 

Regarding the copyright issue of figure one (partial distribution of the schools) study area, we have taken GPS coordinates (both altitude and latitude) of each school, zones, and towns during the data collection. We used QGIS software, version 3.16.16 Hannover, to merge both administrative areas and the location of schools. The shape file we used to locate the Ethiopia boundary is from Central Statical Authority 2019, which is available to use for the public good. We included the software developers for using freely accessible QGIS software. We believe the research team compiled and organized the map for the purpose of this study, not directly taken anywhere. Further explanations were included in the main manuscript under the methods section.

---

## [Decision Letter · Decision Letter 1]

10 Jan 2023

Escalating spread of SARS-CoV-2 infection after school reopening among students in hotspot districts of Oromia Region in Ethiopia: Longitudinal study

PONE-D-22-12485R1

Dear Dr. Gobena,

We’re pleased to inform you that your manuscript has been judged scientifically suitable for publication and will be formally accepted for publication once it meets all outstanding technical requirements.

Kind regards,

Md. Tanvir Hossain

Academic Editor

PLOS ONE

Reviewers' comments:

Reviewer's Responses to Questions

**Comments to the Author**

1. If the authors have adequately addressed your comments raised in a previous round of review and you feel that this manuscript is now acceptable for publication, you may indicate that here to bypass the “Comments to the Author” section, enter your conflict of interest statement in the “Confidential to Editor” section, and submit your "Accept" recommendation.

Reviewer #1: All comments have been addressed

2. Is the manuscript technically sound, and do the data support the conclusions?

Reviewer #1: Yes

3. Has the statistical analysis been performed appropriately and rigorously? 

Reviewer #1: Yes

4. Have the authors made all data underlying the findings in their manuscript fully available?

Reviewer #1: (No Response)

5. Is the manuscript presented in an intelligible fashion and written in standard English?

Reviewer #1: Yes

6. Review Comments to the Author

Reviewer #1: (No Response)

7. PLOS authors have the option to publish the peer review history of their article (what does this mean?). If published, this will include your full peer review and any attached files.

Reviewer #1: No
